# RNA Helicases as Shadow Modulators of Cell Cycle Progression

**DOI:** 10.3390/ijms22062984

**Published:** 2021-03-15

**Authors:** Olga Sergeeva, Timofei Zatsepin

**Affiliations:** 1Skolkovo Institute of Science and Technology, Bolshoy Boulevard 30b1, 121205 Moscow, Russia; T.Zatsepin@skoltech.ru; 2Department of Chemistry, Lomonosov Moscow State University, 119992 Moscow, Russia

**Keywords:** RNA helicases, cell cycle, regulation

## Abstract

The progress of the cell cycle is directly regulated by modulation of cyclins and cyclin-dependent kinases. However, many proteins that control DNA replication, RNA transcription and the synthesis and degradation of proteins can manage the activity or levels of master cell cycle regulators. Among them, RNA helicases are key participants in RNA metabolism involved in the global or specific tuning of cell cycle regulators at the level of transcription and translation. Several RNA helicases have been recently evaluated as promising therapeutic targets, including eIF4A, DDX3 and DDX5. However, targeting RNA helicases can result in side effects due to the influence on the cell cycle. In this review, we discuss direct and indirect participation of RNA helicases in the regulation of the cell cycle in order to draw attention to downstream events that may occur after suppression or inhibition of RNA helicases.

## 1. Introduction

Cell division can be formally presented as a series of coordinated events that form the cell cycle. The transition of the cell cycle through all phases is determined by the interaction of cyclin-dependent kinases (CDK) and CDK-bound proteins with cyclins [1]. The interactions of certain CDKs and cyclins and a detailed description of the functions of each pair in cell cycle transitions have been summarized in recent reviews [2,3,4]. Complex regulation of these key proteins occurs at all stages of their life cycle—transcription, translation, post-translational modification and proteolytic degradation mediated by ubiquitin. Thus, understanding the entire regulatory network of the cell cycle and other proteins that are involved in the regulation of key events in this process, including protein kinases and phosphatases [5], transcription [6] and translation factors [7], is important for accurate control of cells with a normal or dysregulated cell cycle.

Disturbances in the cell cycle can result in the development or enhancement of many pathophysiological conditions—cancer, ischemia/reperfusion injury, atherosclerosis, inflammation and neurodegeneration [8]. Various cell cycle regulators were considered as attractive targets for therapy [9,10]. The first generation of cell cycle inhibitors were the so-called pan-CDK inhibitors, including flavopiridol, (R)-roscovitine and olomoucine [11]. Then, compounds selective for CDK4/6—palbociclib, ribociclib and abemaciclib—revolutionized the clinical management of hormone receptor positive metastatic breast cancer [12]. Among the key limitations for wider use of CDK inhibitors are poor targeted delivery into tumor cells and low targeting selectivity to various cell types, which leads to off-target toxicity. Thus, inhibitors with more precise targeting to other cell cycle proteins have been tested in preclinical and clinical studies (MK-8776 and LY2606368 for check point kinase CHK1 and AZD1775 for WEE1) [11]. However, they still demonstrate a fairly low therapeutic index. Thus, targeting the upstream regulators of cyclins/CDK and other key proteins in the cell cycle would appear to be a promising therapeutic approach.

There are numerous reports indicating that RNA helicases are involved in the regulation of transcription, translation, splicing and RNA transport. Thus, the participation of RNA helicases in the main processes of RNA metabolism makes them important indirect regulators of the cell cycle [13]. RNA helicases can control the levels of the main regulators of the cell cycle—cyclin-dependent kinases, cyclins and their inhibitors—mainly at the stage of translation initiation [14]. RNA helicases have already been evaluated as targets for antiviral and anticancer therapy [15,16,17,18,19]. However, targeting multifunctional proteins such as RNA helicases can lead to undesirable effects by disrupting downstream processes, such as affecting the activity/levels of cell cycle controllers. In this review, we discuss the role of RNA helicases in the regulation of the cell cycle in order to draw attention to subsequent events that may occur after suppression or inhibition of RNA helicases.

## 2. Basic Regulation of Cell Cycle

The cell cycle is a sequential one-way process that guides the cell through initial growth (G1 phase), DNA replication (S phase), another growth period (G2 phase) and finally segregation of the chromosomes into two new nuclei (M phase), followed by cell division resulting in the birth of daughter cells. These major cell cycle events are triggered by CDKs controlled by cyclin binding and/or phosphorylation. Each stage of the cell cycle is controlled by a set of specific CDKs and cyclins (Figure 1A). CDK1 forms complexes with mitotic cyclins (cyclin A and cyclin B) and launches cells into mitosis [20]. Degradation of cyclin B turns off CDK1 and allows the cell to exit mitosis [21]. CDK2 is an important regulator of both G1 and S phases and interacts mainly with cyclin A and cyclin E [22]. CDK3 is involved in the G0—G1 and G1—S cell cycle transitions [23,24,25,26]. CDK4 and CDK6 interact with D-type cyclins (D1, D2 and D3) and are active in G1 phase before the contribution of cyclin E–CDK2 [24]. Their transcription is stimulated and repressed by several transcription factors, including B-MYB, E2F, FOXM1 and NF-Y [27]. Cell cycle progression is also under control of negative regulators, such as CDK inhibitors—INK4 and Cip/Kip protein families [28,29]. The INK4 protein family (p16, p15, p18 and p19 proteins) targets CDK4(6)/cyclin D complexes, while the Cip/Kip protein family (p21, p27 and p57 proteins) targets the CDK2/cyclin E complex [30]. The amount of each cyclin during the cell cycle is regulated by modulation of transcription and/or protein degradation. For example, mRNAs of mitotic cyclins behave similarly to proteins—they are upregulated during the G2 phase and decreased after mitosis [27] due to the incapsulation in the stress granules [28]. The tumor suppressor protein Rb also regulates entry into the cell cycle and G1/S progression by binding to the transcription factor family E2F, followed by repression of genes specific to the cell cycle [31]. The PPTG protein interacts with cell cycle proteins in G1/S phase and is associated with chromosomal instability [32]. The tumor suppressor protein p53 also plays an important role in the arrest of the cell cycle at checkpoints G1 and G2. p53 can activate the transcription of p21, an inhibitor of cyclin-dependent kinase, which blocks the activation of various G1 cyclin/cyclin-dependent kinase complexes [33]. There are also alternative ways of indirect regulation: active cyclin–CDK pairs can be inactivated by small inhibitory proteins CKI [25]. Ubiquitin-mediated proteolysis of CKI and other proteins that interact with CDK and cyclins is also crucial for cell cycle transitions [26]. Some protein kinases are classified as CDKs only because they have high homology with other CDKs, especially within the kinase domain. Hence, not all cyclin partners have been found yet, while some cyclins and CDKs have regulatory functions unrelated to cell cycle.

However, the cell cycle is not just a mechanical automated cycle—before moving to the next phase, the cell must go through checkpoints to ensure that each stage is fully completed without any defects. In general, checkpoints include a sensor that monitors defects at the stage of the cell cycle, signal transducers and an effector that inhibits the transition of the cell cycle if something goes wrong [3,4,22]. During each major checkpoint, DNA damage is also monitored followed by obligatory DNA repair before cell cycle progression [33].

## 3. RNA Helicases in the Regulation of Cell Cycle

RNA helicases unwind duplexes and stem-loops in RNA, thereby introducing structural changes into the RNA and RNA–protein complexes and switching their activity using NTP as an energy source [34,35]. RNA helicases are involved in all aspects of RNA metabolism: ribosome biogenesis (DDX10, DDX17) [36], pre-mRNA splicing (DDX41, DDX48), mRNA translation (DDX3, DHX29), nuclear export (DDX39, DDX45), mRNA decay (DDX6), miRNA-induced gene silencing (DDX5, DHX9) and mRNA transportation and storage (DDX4, DDX3) [37]. Despite these basic functions, RNA helicases contribute to the post-translational modification of proteins and the regulation of signaling pathways in the cell [38,39]. Also, RNA viruses can hijack RNA helicases in the host cell to maintain their life cycle [40]. Therefore, dysregulation of the expression or activity of these proteins and mutations in the coding region of genes leads to the development or progression of many diseases. Some small molecules that target RNA helicases—inhibitors of eIF4A, like rocaglamide and silvestrol [41], and DDX3 inhibitor RK-33 [42]—have already reached preclinical trials for cancer treatment. Several candidate antivirals have been developed that target RNA helicases (e.g., thiazolyl phenyl compounds that disturb the life cycle of herpes simplex viruses [43]). Here, we will briefly highlight the main functions of some RNA helicases and then focus on their role in the regulation of the cell cycle (Table 1), since dysregulation of the cell cycle can be either beneficial or negative for the pharmacological effect of the suppression or inhibition of RNA helicases.

RNA helicases belong to the main superfamilies SF1 and SF2, which are also subdivided into families based on their motif composition. The catalytic cores of SF1 and SF2 helicases share almost identical folds and significant structural similarity. The presence or absence of a specific motif determines the functions of RNA helicases. The central structural element of SF2 superfamily is formed by two RecA-like domains carrying conservative motifs important for these RNA helicases. Motifs I, II, V and VI are required for binding and hydrolysis of a nucleoside triphosphate; motifs Ia, Ib and IV are involved in RNA binding; and the motif III participates in coupling of ATPase and unwinding activities (Figure 1B). In the DEAD-box proteins, motif II includes the sequence D-E-A-D, which is the origin of their name. In humans, the largest families of RNA helicases are the DEAD box (DDX; 40 members) and DEAH box (DHX; 15 members) proteins [36,44,45,46].

The main role of DDX3 helicase from the DExD protein family is the regulation of translation. DDX3 represses cap-dependent translation by trapping eIF4E into a translationally inactive complex [47,48] or binds to eIF4E along with several translation initiation factors, including eIF4A, eIF4G, eIF2A, eIF3 and poly(A)-binding protein (PABP), and facilitates translation of mRNA containing a structured 5′ untranslated region [49], or it can interact with eIF3 [50]. In addition to participating in translation, DDX3 is a multifunctional protein that is involved in almost all aspects of RNA metabolism—transcription, splicing, translation and RNA decay. During early embryonic development in mice, DDX3 also regulates cell survival by adjusting p53-induced apoptosis [51]. In the cell cycle, DDX3 participates in the transition between the G1 and S phases of the cell cycle, regulating the initiation of translation of cyclin E1 [52]. DDX3 directly interacts with pre-mRNA of transcription factor KLF4 and regulates its splicing. This event ultimately affects the gene expression of the key cell cycle regulators: cyclin A1 and cyclin-dependent kinase 2, which leads to the arrest of the cell cycle in the G1 phase [53]. On the other hand, DDX3 can modulate the transcriptional activity of p21 waf1/cip1 promoter and regulate this cyclin-dependent kinase inhibitor. Thus, DDX3 inhibitors can affect not only the translation of certain (or all [54]) mRNA, but also the cell cycle by changing the expression of key regulators. Recently developed cancer-related DDX3 inhibitors show a synergistic effect with multiple actions—they simultaneously decrease translation and slow down the cell cycle, which aids in the cancer treatment.

Helicase DDX46 is involved in pre-mRNA splicing [55] and innate antiviral response by recruiting the m6A “eraser” ALKBH5 [56]. In addition, inhibition of DDX46 causes cell arrest in G1 phase and apoptosis via phosphorylation of Akt1 protein kinase and IkBa inhibitor [56,57]. Thus, the effect of DDX46 inhibitors may result from the combined effect of PI3K/Akt downregulation and changes in the cell cycle.

DDX6 interacts with several protein complexes and regulates the mRNA life cycle and translation rate. DDX6 can interact with decapping (DCP1A, EDC3, EDC4, Lsm1 and Pat1) and translational machinery, such as eIF4E; translational repression factors (4E-T, ataxin 2/2L and LSM14); and other RNA-binding proteins (RBPs), such as YBX1, IGF2BP2, FXR1, polyA-binding and ribosomal proteins. The association of DDX6 with EDC3 reduces stability of KLF4 mRNA [58], an essential transcriptional factor that regulates the expression of cyclin A1 and CDK2. In the cytoplasm, DDX6 is enriched in P bodies and stress granules and is important for their homeostasis [59,60]. DDX6 also binds to nascent ribonucleoprotein (RNP) transcripts and accompanies the export of maternal mRNA to the cytoplasm as mRNP storage particles [61]. Several studies of human DDX6 and the yeast homolog Dhh1p have shown that these proteins are positive regulators of cell cycle progression. Dhh1p is important for recovery from DNA-damage-dependent G1/S cell cycle arrest [62]. Moreover, overexpression of Dhh1p inhibits cell growth in yeast, probably due to a general repression of translation [63]. RNAi-mediated knockdown of DDX6 leads to cell cycle arrest in the S phase [12]. In addition, DDX6 depletion results in reduced cell viability, increased portion of cells in S phase, increased apoptosis and decreased ability to form tumors in xenograft models [64]. The probable mechanism is based on the regulation of the transcription factor Tcf with subsequent changes in known targets of Wnt/β-catenin pathway, such as cox-2, cyclin D1 and survivin [65]. In addition, overexpression of DDX6 activates the c-Myc oncogene, which is consistent with the results on cell proliferation [66]. Taken together, these findings suggest that DDX6 plays an important evolutionarily conserved role in cell cycle progression and proliferation, probably through the regulation of translation of specific key mRNAs. However, another group showed that overexpression of DDX6 leads to inhibition of cell growth and a decrease in anchorage-independent growth, a hallmark of malignant transformation [67]. Since DDX6 may become a valuable target for cancer therapy [68,69], the significant role of this helicase in the regulation of the cell cycle should be taken into account in order to avoid secondary side effects.

Helicase DDX21 is involved in the processing of ribosomal RNA through interaction with 45S and 32S precursors and regulates 18S and 28S rRNA levels in the cell [70,71]. DDX21 also activates transcription via polymerase I and participates in small nucleolar RNPs (snoRNP)-dependent modification of rRNA and promotes RNA elongation by polymerase II, facilitating the release of P-TEFb from the 7SK snRNP complex and phosphorylation of the C-terminal domain (CTD) of Pol II [72]. DDX21 increases the proliferation of breast cancer cells by activating the AP-1 transcription factor and rRNA processing [73]. Upregulation of DDX21 helicase correlates with the increased number of cells in the S phase and cell proliferation in gastric cancer cells. The proposed mechanism includes the activation of c-Jun transcription, a subunit of the AP-1 transcription factor, crucial for the synthesis of cyclin D1 mRNA and a receptor for activation of kinase C1 [74]. DDX21 is activated by ADP-ribosylation with PARP-1 [75], and therefore PARP inhibitors can be used to indirectly suppress DDX21 and reduce cell proliferation even in cancer cells without defects in DNA repair machinery.

DDX51 binds to pre-60S subunit complexes and facilitates the displacement of U8 snoRNA from pre-rRNA, which is necessary for the removal of the 3′ external transcribed spacer from 28S rRNA and productive downstream processing [76]. DDX51 is a negative regulator of the apoptotic effector p53, and, thus, actively promotes cell proliferation [77]. DDX51 also promotes the proliferation of breast cancer cells by activating the Wnt/β-catenin signaling pathway and affects expression of cyclin D1 [78]. A decrease in DDX51 results in cell cycle arrest in the S phase [79], probably due to the regulation of cell cycle progression via multiple pathways or an alternative function of DDX51 RNA helicase participation in rRNA processing.

DDX5 (also known as p68) is one of the prototypic members of the DEAD box family of RNA helicases. DDX5 and related DDX17 (p72) are involved in a variety of cellular processes, including transcription, pre-mRNA and rRNA processing, alternative splicing and miRNA processing, and they are also deregulated in a range of cancers [80,81,82,83]. It has been shown that DDX5 participates in the replication of the HIV-1 virus [84]. All these functions make DDX5 a prospective target for the treatment of cancer and viral infections [84]. DDX5 plays a proliferative or oncogenic role in cancer through the coactivation of Androgen Receptor (AR) [85], Runx2 [86] and the p50 subunit of NF-κB [87], and upregulation of cyclin D1 and c-Myc consistent with β-catenin activation [88,89] as well as genes necessary for DNA replication. In colon cancer, DDX5 may affect β-catenin in two ways: by protecting β-catenin in the cytoplasm from degradation via dissociation from the cytoplasmic APC/axin/GSK-3β complex or by augmenting β-catenin transcriptional activity in the nucleus [90]. DDX5 is a negative regulator of Wnt signaling and hepatocyte reprogramming in HCC [91]. However, in colorectal cancer cells, DDX5 interacts with noncoding RNA NEAT1, which improves its stability, and sequentially activates Wnt signaling [92]. Moreover, DDX5 was found to activate the transcription of the *Snail1* gene by displacing histone deacetylase from the Snail1 promoter [89], which is consistent with its participation in the epithelial–mesenchymal transition (EMT). In spermatogonia loss of DDX5, cell cycle arrest occurs at both G1/S and G2/M stages. Several cell-cycle-related genes are aberrantly expressed, including strong upregulation of *Cdkn1a* (*p21*). DDX5 also binds to a number of cyclin mRNA transcripts and influences its nuclear export and stability [93]. DDX5 interacts with the early S-phase-upregulated noncoding RNA SUNO1, which promotes the association with RNA pol II on chromatin, thereby promoting transcription of cell cycle genes such as *WTIP* [94]. On the other hand, for DDX5 and DDX17, antiproliferative or tumor suppressive functions are also implied. A role in differentiation has been confirmed by the finding that DDX5 and DDX17 coactivate the myogenic regulatory factor MyoD and are required for differentiation of skeletal muscle cells [95]. DDX5 coactivates the p53 tumor suppressor and is required for a p53-dependent DNA damage response [96,97]. However, a recent study demonstrated that although DDX5 is required for p53-dependent CDKN1 induction and cell cycle arrest, DDX5 is not involved in the induction of apoptosis [97], suggesting that DDX5 may play a role in cell survival in some cases. Moreover, the finding that DDX5 induces the expression of the cell cycle arrest gene *CDKN1*, and, conversely, *cyclin D1* in a different context [98], suggests that DDX5 may have opposite effects on the cell cycle progression under different conditions. DDX5 is required for the progression of G1–S phase in the breast cancer cell line by involvement in the transcriptional regulation of Cdc45/Mcm2-7/GINS complex. DDX5 contributes toward initiation of replication and thus S-phase entry, where it promotes DNA replication preinitiation complex assembly on chromatin [99]. Thus, DDX5 is a viable candidate drug target for selective anticancer therapy directed at those tumors that have an amplified *DDX5* locus. In addition, this data supports the idea that DDX5 is highly context sensitive presumably due to post-translational modifications or specific rules. Thus, DDX5 may not be an effective universal therapeutic target. Furthermore, simultaneous delivery of inhibitors to healthy cell can induce toxic or off-target effects.

DHX33 interacts with the architectural protein UBF and indirectly regulates RNA polymerase I-mediated transcription and rRNA synthesis [100]; promotes the assembly of 80S ribosome at the late stage of mRNA translation initiation [101]; selectively regulates transcription of *MMP9*, *MMP14* and *PLAU* genes involved in the regulation of cancer cell invasion and migration [102]; and participates in the innate immune response to double-stranded and bacterial RNA in the cell cytosol [103]. In addition, DHX33 recruits active RNA polymerase II to facilitate the transcription of many genes associated with the cell cycle: *cyclin A2*, *cyclin B2*, *cyclin E2*, *MCM4*, *MCM7*, *cdc6*, *cdc20* and *E2F1*. Expression of DHX33 is necessary for continuous cell proliferation as it promotes cell cycle progression at the G1/S, G2/M and metaphase–anaphase transitions [104]. DHX33 is induced by PI3K and mTOR inhibitors and contributes to the development of glioblastoma by accelerating the cell cycle [105]. In addition, DHX33 interacts with transcriptional factor AP-2β and binds to the promoters of the genes involved in the cell cycle processes: *MCM2*, *MCM4*, *CDC26*, *CCNB2*, *CCNE1*, *CCNE2* and *CCND3* [106]. DHX33 participates in Ras-driven lung cancer development by regulation of certain genes associated with cell proliferation, such as *cdc6*, *cyclin D1* and *Ki-67*. Thus, DHX33 contributes to the regulation of various aspects of cell proliferation and migration during cancer development. Hence, the exact molecular mechanism remains to be revealed [107].

DDX41 acts as an intracellular DNA sensor in myeloid dendritic cells via the STING-TBK1-IRF3 pathway [108]. Besides its role in innate immunity, DDX41 is also associated with hereditary diseases. Mutations in the DDX41 gene lead to loss of tumor suppressor function due to altered pre-mRNA splicing and RNA processing [109]. Also, helicase DDX41 is a new repressor of one of the most studied inhibitors of CDK1A, p21WAF1/CIP1, which makes DDX41 an important participant in the regulation of the cell cycle. DDX41 reduces translation of p21 mRNA by binding to the 3′-UTR [110]. 

Helicase eIF4A is a canonical protein of the translation initiation process that unwinds long and complex secondary structures in the 5′-UTR, which are common for many eukaryotic mRNAs [111]. In addition to its helicase activity, eIF4A may function as a regulatory switch to control the conformation of the 43S preinitiation complex during mRNA recruitment in cap-dependent translation [112]. In human breast adenocarcinoma cells, inhibition of eIF4A blocks cell cycle progression at the G1/S phase transition, likely through loss of Cyclin D1, Cyclin D2 and Cdk6, and induces apoptosis, in part, by inhibiting translation of BCL2. However, the main influence of eIF4A on cell cycle progression is contributed at the level of the translation initiation process [113]. Also, eIF4A is crucial for the translation of viral IRES mRNAs [114]. Due to the important role of eIF4A in the development of cancer and viral diseases, many inhibitors have been developed to date [115]. Nevertheless, direct targeting of this essential protein can lead to strong adverse effects.

DHX9 mediates binding of the epidermal growth factor receptor (EGFR) with the cyclin D1 promoter, which stimulates its transcription activation [116]. DHX9 also forms a complex with RNA Pol II and EWS–FLI1 to augment EWS–FLI1-dependent CCND1 transcription [117]. Additionally, DHX9 can interact with noncoding RNA pncCCND1_B and repress cyclin D1 transcription, exhibiting both oncogenic and tumor suppressive functions [118]. DHX9 can also interact with p16INK4a promoter and activate its transcription, which leads to the inhibition of cyclin D1–CDK4/6 complex formation, thus promoting cell cycle regulation [119]. DHX9 is also capable of suppressing activity of the tumor suppressor BRCA1, which functions in the DNA damage response and cell cycle arrest [120]. DHX9 is upregulated in many cancers, and its downregulation causes p53-mediated apoptosis, which also promotes cell cycle regulation [121]. It was demonstrated that DHX9 interacts with CIP1-interacting zinc finger protein 1 (CIZ1) and contributes to CIZ1 nucleolar localization in S phase. Nucleolar CIZ1-DHX9 localization is required for efficient cell cycle progression and contributes to noncanonical roles in the nucleolus [122]. DHX9′s multitude of functions in the development of cancer highlight a pivotal role in malignancy and the potential as both a biomarker and selective target for cancer therapy.

RNA helicase DDX56 is overexpressed in different cancer types (e.g., in colorectal cancer) and leads to a poor prognosis. DDX56 can promote cell proliferation by inducing oncogenic splicing alteration in a cell cycle checkpoint gene, *WEE1,* a G2–M cell cycle checkpoint. Hence, the molecular mechanism of *WEE1* abnormal splicing of regulation by DDX56 remains unknown [123]. Inhibition of DDX56 in osteosarcoma cells also decreases cell proliferation and promotes p53-mediated apoptosis [124]. 

RNA helicase UAP56/DDX39B participates in the resolution of the nonscheduled R loops, representing a major source of DNA damage and replication stress in human cells. UAP56 depletion causes replication fork stalling and leads to a significant increase in the percentage of damaged cells in all phases of the cell cycle but has a major impact in G1 cells [125].

We also want to mention a set of helicases that can indirectly influence the cell cycle. Among them is helicase SUV 3, which is capable of unwinding the secondary structures of RNA and DNA. This helicase is localized in the mitochondria and is part of the mitochondrial degradosome. A decrease in the amount of SUV 3 leads to the accumulation of nondegraded mitochondrial RNA and, as a consequence, to changes in the cell cycle [126]. In embryonic stem cells, DDX18 helicase counteracts PRC2 to ensure chromatin availability of rDNA genes and, therefore, promotes high levels of rRNA transcription, ribosome biogenesis and translation, which are required for self-renewal of ESCs and, therefore, are required for cell cycle progression [127]. In addition, mutations in this putative DExH box RNA helicase lead to a delay in G2/M due to the influence on the splicing of pre-mRNA of the cell cycle regulators [128]. The level of RNA helicase HELZ affects phosphorylation of the ribosomal protein S6 and the number of polysomes in the cell, which in turn causes changes in the cell cycle [129]. 

## 4. Conclusions

RNA helicases are highly conserved enzymes crucial for RNA metabolism. Most of the RNA helicases discussed here have become [133] or may become promising biomarkers or targets for the diagnosis, prognosis and treatment of viral diseases and cancers [134,135]. Small molecule inhibitors have already been developed and validated for the canonical translation initiation factor eIF4A: hypericin [136], hippuristanol [137], pateamine A [138], rocaglamides [139] and DDX3 (RK-33 [140]) and DDX5 (RX-5902 [141]) helicases. Among them, we want to highlight DDX3 as a confirmed target for cancer and antiviral therapy [142], while others, like DDX46, DHX9 and DDX5, are still poorly studied in this context. DDX5 and DDX6 are involved in replication of HIV and hepatitis C viruses [84,143], while DDX21 contributes to the life cycle of some RNA viruses such as cytomegalovirus, influenza A virus and retroviruses through various mechanisms [144,145,146]. Thus, all these helicases can become potential targets for antiviral therapy.

However, RNA helicases are involved not only in the regulation of RNA metabolism. They also participate in various aspects of cell cycle regulation (Figure 2 and Table 1). In some cases, the effect on the cell cycle is just a result of downstream effects, while there are many examples when RNA helicases exhibit moonlight functions and directly regulate the expression of master cell cycle regulators. RNA helicases can influence cell cycle progression not only at the checkpoint stages, but also at each phase using three main mechanisms of action. Most of RNA helicases mentioned in this review (DDX6, DDX21, DDX51, DDX5, DX17, DHX33, DHX9, DDX56) are involved in the regulation of pre-mRNA transcription or splicing of some cell cycle regulators, such as cyclins and cyclin-dependent kinases (Table 1). Another mechanism of the cell cycle regulation by RNA helicases is based on the changes in translation of cellular regulators. DDX3, eIF4A and DDX41 regulate the stage of translation initiation of cyclins and cyclin-dependent kinases and tune their levels at different steps (Table 1). DDX46 and DDX18 use a third approach—they are involved in the regulation of post-translational and epigenetic modification of the effectors involved in signaling pathways that regulate the cell cycle progression (Table 1).

Thus, inhibition or downregulation of RNA helicases, discussed above as antiviral and anticancer targets, will also lead to a significant effect on the cell cycle. Inhibition of DDX3 causes a decrease in the amount of cyclin E1, A1 and CDK2 and stops the progress of cell cycle [39,52,53,147]. A decrease in DDX46 affects cell recovery from DNA-damage-dependent G1/S cell cycle arrest [57]. DDX21 is important for the expression of cyclin D1 and activation of C1 protein kinase. Therefore, inhibition of DDX21 causes cell cycle arrest [73,74,145]. There are contradictory data on DDX5—on the one hand, DDX5 is necessary for the synthesis of cyclin D1 and c-myc. On the other hand, DDX5 is important for p53 activation [85,86]. Thus, the effect on cell cycle can be cell-type dependent. DHX33 is a regulator of transcription of many cyclins and CDKs—therefore, its inhibition causes stagnation of the cell cycle [101,102,103,104,105,106,107]. DDX41 is a repressor of CDK inhibitor, which is also important for cell cycle progression [108,109,110]. DHX9 regulates cyclin D1–CDK4/6 complex formation [118]. DDX56 is a regulator of the G2–M cell cycle checkpoint gene, *WEE1* [123]. All of these events should be taken into account, as sometimes these phenotypes can be detrimental for research or drug development. Inhibition or downregulation of RNA helicases may cause cell cycle arrest, which leads to changes in the given transcriptome and proteome data. In addition, RNA helicases (for example, DDX5 and DDX17 [148]) are involved in the transcriptional regulation of long-noncoding RNA (lncRNA), which is crucial for physiological and pathophysiological conditions [149,150]. Downregulation of lncRNAs by small molecules or oligonucleotides [151,152] can also affect the RNA helicase functions [139] and, finally, cell cycle progression [153]. 

Multiple functions, sometimes with the opposite direction depending on the type of cells or conditions, make some RNA helicases potential biomarkers and selective targets for therapy. Moreover, their contribution to various processes may depend on the level of the protein in the cell and/or on the inhibited protein domain responsible either for ATP- or RNA-binding. Here, we reviewed the effects of RNA helicases on cell cycle and cell proliferation. These follow-up events may be crucial for development of therapeutics due to potential off-target effects. Problems with pharmacological intervention can also be resolved by targeted delivery of inhibitors to cancer or infected cells [153]. On the other hand, thorough evaluation of the RNA helicases together with the inhibitor/suppressor in vitro and in vivo should be sufficient for successful preclinical development without unexpected toxicity or side effects in long-term treatment in later stages of clinical trials. Nevertheless, we expect that pharmacological intervention to the processes driven by RNA helicases can be beneficial in many cases and various inhibitors will become approved drugs.

## Figures and Tables

**Figure 1 ijms-22-02984-f001:**
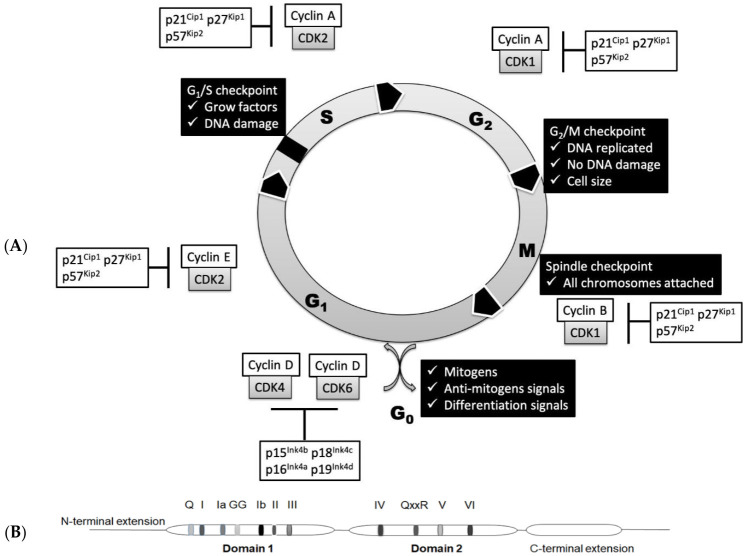
(**A**) Regulation of the cell cycle in mammals. Contribution of cyclin-dependent kinases (CDKs), cyclins and CDK inhibitors at each phase. (**B**) Schematic depiction of domains and motifs of SF2 RNA helicases based on the Mss116p structure.

**Figure 2 ijms-22-02984-f002:**
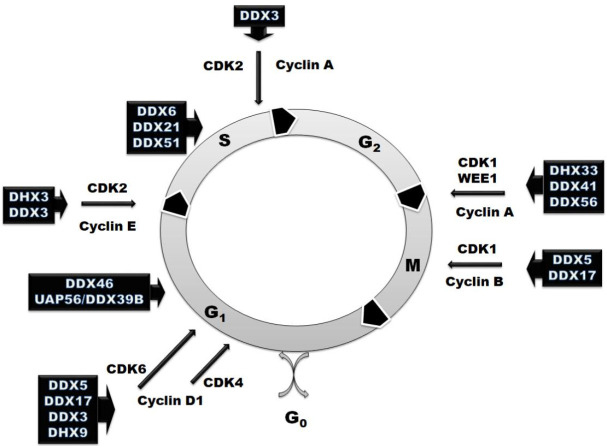
Schematic representation of the involvement of RNA helicases in the cell cycle regulation with marked cyclin-dependent kinases (CDKs) and their regulatory partner proteins, the cyclins based on the analyzed published data represented in Table 1.

**Table 1 ijms-22-02984-t001:** Summary of the RNA helicases involvement in the cell cycle regulation.

Helicase	Target Crucial for the Cell Cycle	Cell Cycle Phase or Transition
DDX3	cyclin E1, cyclin A1, cyclin D1, CDK2	G1–S [39,52,53,130]
DDX46	phosphorylation of Akt1 and IkBa inhibitor	Arrest at G1 phase [57]
DDX6	transcription factor Tcf, target genes of Wnt/β-catenin—c-Myc, cyclin D1, cox-2, livin, survivin and VEGF	Arrest at S phase [130,131]
DDX21	c-Jun, required for the synthesis of cyclin D1	Arrest at S phase [73,74]
DDX51	TGFBR2	Arrest at S phase [132]
DDX5	cyclin D1, c-Myc, β-catenin activation, Cdc45/Mcm2-7/GINS	Arrest at G1 and M phases [85,86,92,99]
DDX41	p21WAF1/CIP1	G2–M [16,110]
DHX33	cyclin A2, cyclin B2, cyclin E2, MCM4, MCM7, cdc6, cdc20, and E2F1, cdc6, cyclin D1 and Ki-67	G1–S; G2–M [104,105,106,107]
eIF4A	cyclin D1, cyclin D2, CDK6, CDK8 and Bcl2	All stages [111,113,122]
DHX9	Cyclin D1, zinc finger protein 1	G1–S [117,118,119,120,121,122]
DDX56	WEE1	G2–M [123,124]
UAP56/DDX39B	R-loop	Arrest at G1 phase [125]

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
