# Peer review of "RNA Helicases as Shadow Modulators of Cell Cycle Progression"

_ijms, 2021, doi:10.3390/ijms22062984_

Round 1

Reviewer 1 Report

The Review article entitled: “RNA helicases as shadow modulators of cell cycle progression” attempts to link the function of this large family of RNA helicases to the cell cycle. However, as presented, the review overall, did not generate a compelling case in support of such a connection.

Another shortcoming of the review is that it does not include recent studies providing more direct link to cell cycle regulation. For example, see the recent studies: PMID: 33093612; PMID: 33108271. Very few 2019 and 2020 articles are being referenced.

The review contains errors: for example line 222 describing the study by Nichols et al, Ref#92.

The review lacks critical evaluation of the referenced studies; for example study #94. Is there any follow-up of this observation?

Fig. 2 should list the references providing the critical evidence for assigning the indicated helicases to specific cell cycle phases.

Line 216 must be referenced and must be expanded to describe the studies that support the tumor suppressor function of DDX5. The same effort should be made towards the role of DDX5 in Wnt/beta catenin signaling and activation.It is important to emphasize the context-dependent function of this intriguing helicase.

The DEAD box RNA helicases are ATPases. This correction should be made.-line 127

A diagram should be included –paragraph starting line120- to illustrate the organization /motif composition of SF1 and SF2 helicases.

Author Response

Thank you for your time to evaluate the reviewed and give us many valuable comments and suggestions. We have seriously considered an addressed all of your recommendations as explained below. We reanalyze the published papers from 2019 and 2020 and expanded our review. All corrections are indicated in yellow.

The Review article entitled: “RNA helicases as shadow modulators of cell cycle progression” attempts to link the function of this large family of RNA helicases to the cell cycle. However, as presented, the review overall, did not generate a compelling case in support of such a connection.

Another shortcoming of the review is that it does not include recent studies providing more direct link to cell cycle regulation. For example, see the recent studies: PMID: 33093612; PMID: 33108271. Very few 2019 and 2020 articles are being referenced. - Thank you for bring to our attention to this point. We have added the recommended research items in our review. In particular the references below were added:

doi: 10.1111/cas.14601, doi: 10.1128/MCB.00460-19,

doi:10.1073/pnas.1000743107, doi: 10.18632/oncotarget.5033,

doi:10.1158/0008-5472.CAN-182403, doi:10.1074/jbc.M004481200,

doi:10.1038/sj.onc.1206195, doi:10.1038/onc.2016.52,

doi: 10.1111/cas.14163, doi:10.3389/fbioe.2020.00588,

doi:10.1101/gad.336024.119, doi: 10.1038/s41598-020-75160-z,

doi:10.7554/eLife.55102.

The review contains errors: for example line 222 describing the study by Nichols et al, Ref#92. we corrected this error and checked all references by eyes.

The review lacks critical evaluation of the referenced studies; for example study #94. Is there any follow-up of this observation? – we added explanation in the text.

Fig. 2 should list the references providing the critical evidence for assigning the indicated helicases to specific cell cycle phases. – Fig. 2 was proposed based on the data in Table 1. We marked it in the text.

Line 216 must be referenced and must be expanded to describe the studies that support the tumor suppressor function of DDX5. The same effort should be made towards the role of DDX5 in Wnt/beta catenin signaling and activation.It is important to emphasize the context-dependent function of this intriguing helicase. - we added explanation in the text.

The DEAD box RNA helicases are ATPases. This correction should be made-line 127 – we corrected this point.

A diagram should be included –paragraph starting line120- to illustrate the organization /motif composition of SF1 and SF2 helicases. –we added the diagram.

Reviewer 2 Report

Overall, this is a nice review coupling RNA helicases and cell cycle progression.  Essentially all of my comments are strictly on the English used in a few places and these are indicated below:

line 44 should read "...cell cycle would appear to be a..."

line 114 should read "inhibitor RK33 [42} have already..."

line 133 should read "...including eIF4A, eIF4G, eIF2alpha, eIF3 ..."; note it is not clear if the authors might have meant eIF2A instead of eIF2a.

line 159 should read "...and FXR1, and ply(A)-binding protein ..."

line 164 should read "... DDX6 and the yeast homolog ..."

line 183 should read " and regulates 18S and 28S"

line 193 should read PARP-1 [75], and therefore"

line 212 cyclin D1 and c-Myc consistent"

line 216 should read "is consistent with its participation in..." and define EMT.

line 262 should read " DNA. This helicase is ..."

Table 1 eIF4a should be eIF4.

line 299 should read "cell cycle regulator mRNAs."

line 301 should be eIF4A, not eIF4a.

line 321 should read "involved in the transcriptional regulation of"

It is unclear to this reviewer why the references in general seemed to be numbered twice (14.  [14]).  Second, several references seem to have separated the main reference from the https designation (ref. 100, 110125, 128.  References 70 and 132 are indented and the reference for 136 is incomplete.

Author Response

Thank you for your time to evaluate the reviewed and give us many valuable comments and suggestions. We have corrected all English errors and indicated the corrections with yellow.

Overall, this is a nice review coupling RNA helicases and cell cycle progression.  Essentially all of my comments are strictly on the English used in a few places and these are indicated below:

line 44 should read "...cell cycle would appear to be a..."

line 114 should read "inhibitor RK33 [42} have already..."

line 133 should read "...including eIF4A, eIF4G, eIF2alpha, eIF3 ..."; note it is not clear if the authors might have meant eIF2A instead of eIF2a. Thank you for the comment, something was wrong with the formatting because we also find eIF4a instead of eIF4A in the final version. Of course we wrote about eIF2A because for this protein the direct interaction with eIF4E was demonstrated.

line 159 should read "...and FXR1, and ply(A)-binding protein ..."

line 164 should read "... DDX6 and the yeast homolog ..."

line 183 should read " and regulates 18S and 28S"

line 193 should read PARP-1 [75], and therefore"

line 212 cyclin D1 and c-Myc consistent"

line 216 should read "is consistent with its participation in..." and define EMT.

line 262 should read " DNA. This helicase is ..."

Table 1 eIF4a should be eIF4. –

line 299 should read "cell cycle regulator mRNAs."

line 301 should be eIF4A, not eIF4a.

line 321 should read "involved in the transcriptional regulation of"

It is unclear to this reviewer why the references in general seemed to be numbered twice (14.  [14]).  Second, several references seem to have separated the main reference from the https designation (ref. 100, 110125, 128.  References 70 and 132 are indented and the reference for 136 is incomplete. - Thank you for bring to our attention this part of literature. We have corrected all points in this section.

Round 2

Reviewer 1 Report

The revision are acceptable.